# A Review of Advanced Molecular Engineering Approaches to Enhance the Thermostability of Enzyme Breakers: From Prospective of Upstream Oil and Gas Industry

**DOI:** 10.3390/ijms23031597

**Published:** 2022-01-30

**Authors:** Muhammad Naeem, Amjad Bajes Khalil, Zeeshan Tariq, Mohamed Mahmoud

**Affiliations:** 1Department of Bioengineering, King Fahd University of Petroleum and Minerals (KFUPM), Dhahran 31261, Saudi Arabia; g201908410@kfupm.edu.sa; 2Department of Petroleum Engineering, King Fahd University of Petroleum and Minerals (KFUPM), Dhahran 31261, Saudi Arabia; zeeshan_tariq3@hotmail.com

**Keywords:** enzyme engineering, enzyme breakers, direct evolution, rational design, thermostability

## Abstract

During the fracture stimulation of oil and gas wells, fracturing fluids are used to create fractures and transport the proppant into the fractured reservoirs. The fracturing fluid viscosity is responsible for proppant suspension, the viscosity can be increased through the incorporation of guar polymer and cross-linkers. After the fracturing operation, the fluid viscosity is decreased by breakers for efficient oil and gas recovery. Different types of enzyme breakers have been engineered and employed to reduce the fracturing fluid′s viscosity, but thermal stability remains the major constraint for the use of enzymes. The latest enzyme engineering approaches such as direct evolution and rational design, have great potential to increase the enzyme breakers’ thermostability against high temperatures of reservoirs. In this review article, we have reviewed recently advanced enzyme molecular engineering technologies and how these strategies could be used to enhance the thermostability of enzyme breakers in the upstream oil and gas industry.

## 1. Introduction

Hydraulic fracturing is the technique of fracturing rock through a pressurized fluid to enhance the productivity of hydrocarbons such as oil and gas. The process involves the high-pressure injection of pressurized fluid into the deep rock to create cracks in it through which the flow of oil or gases occurs spontaneously [1]. In this technique, various fracturing fluids are used; the ideal fracturing fluid contains sufficient viscosity which is important to transport the proppants to develop the cracks in the deep rocks. Different polymers are used to suspend proppants. A polymer enhances the fluid viscosity to suspend and initiate fractures. Crosslinkers are used to further enhance the viscosity. The viscosity of fracturing fluid is crucial to developing fracture geometry [2]. After the transfer of proppants, the fluid should lose viscosity for spontaneous flow back of hydrocarbons. Breaking of the fracturing fluid viscosity is very important as it restores the proppant permeability. For that purpose, different breakers are commonly used to reduce viscosity. Normal industry practice is to use enzymes, metal persulfate, oxidizers, and peroxides as breakers [3].

Guar gum and its derivatives are widely used as thickening agents to increase the viscosity of fracturing fluid or proppants, which count as 90%. of seeds of guar plant *Cyanaposis tetragonolobus*. Guar is a non-ionic branch polysaccharide composed of a linear B-D-mannopyranosyl backbone linked to a-D-galactopyranosyl that each exhibits a mannose and galactose ratio of 2:1 [4]. The first patent of guar gum as a viscosity enhancer during the hydraulic fracturing operations came in 1962 [5]; further, the efficiency has been improved through different types of guar mixtures and derivatives [6]. Guar gum has a diverse ability for its chemical modification and crosslinking. Hydrophobically modified guar gum (HMGG), a new type of guar gum produced for fracturing fluid, possesses hydrophobic and hydrophilic parts. HMGG is high viscous due to its excellent surfactant low surface tension and antifoaming characteristics [7,8]. Increasing the viscosity of hydraulic fluid during the fracturing operation is important to generate the cracks as well as reducing the viscosity after fracturing fluid operation is highly crucial for permeability of oil and gas, which is why different types of breakers are used to reduce fracturing fluid’s viscosity, including oxidizers, enzymes, and acids. All these breakers reduce the viscosity of fluid or gel by breaking the guar polymer chains. Oxidizers that decrease the gel’s molecular weight (MW) by degrading the guar polymer backbone are persulfate, peroxidase, bromate, and chlorites. The reduction in MW causes a reduction in the viscosity of the gel. The commonly used oxidizers as viscosity reducers are sodium persulfate, ammonium persulfate, and potassium persulfate [9] In acids as gel breakers, weak acids are used as a breaker including tannic and citric acid. The acids lower the pH of the liquid solution and decrease the viscosity. In one study, tannic acid encapsulation could gradually break the fluid viscosity up to 275 °F. Acid breakers work effectively at higher temperatures compared to oxidizer breakers [10]. Enzyme breakers, first introduced in 1992 for low-temperature and high-temperature fracturing applications, are bioactive polymers of amino acids that can reduce hydraulic fluid viscosity by degrading glycoside bonds of guar-based gel polymers. Enzyme breakers have several advantages compared to traditional chemical oxidizers such as being environmentally friendly, their ability to break long-chain polymers without altering near-wellbore petrophysical impairment, and their effectiveness—a minimal number of enzymes can do a lot of work because of their catalytic nature, etc. [11]. Conventionally, the enzyme breakers α-1,6-galactosidase and β-1,4-mannanase were isolated from the hyperthermophilic bacterium *Thermotoga neapolitana* to degrade guar gum. As compared to oxidizers, enzymes can degrade guar polymers at multiple sites rather than the backbone. Enzyme breakers have several advantages over oxidizers and acid breakers. (1) Enzymes act as a biocatalyst; the enzyme cannot be consumed during the degradation process. (2) Enzymes can completely degrade guar gum. (3) It is believed that there is no undesirable or side reaction in the enzyme gel degradation process. (4) Enzymes can break guar gum into smaller fragments with a small amount of residue. (5) Enzymes are substrate specific that increase their specificity to the target molecule. (6) Enzymes do not affect the tubes or equipment during hydraulic fracturing [11]. Enzyme breakers have been commonly used in a water-based fracturing fluid for many decades. It has been discovered that the enzyme break time is a function of enzyme concentration, breaker type, and polymer loading, and enzyme breakers take more time than oxidative breakers [12].

However, the application of enzymes as gel breakers in the upstream oil and gas industry is limited due to thermostability issues. Many enzymes cannot work under extreme temperatures of oil wells or reservoirs. The purpose of this review article is to briefly overview the most recent enzyme engineering techniques (Figure 1) such as direct evolution and rational design, and how these engineering techniques can be employed to increase the thermostability of enzyme breakers to work under extreme temperatures of deep rock cracks to induce the permeability of oil and gas.

## 2. Enzyme Thermostability Issues in Upstream Oil and Gas Applications

In the oil and gas industry, enzyme thermostability is very crucial during drilling and completion operations. Most of the enzymes degrade above 200 °F leading to failure of the anticipated operations. Polymers are used in drilling and completion fluids to enhance the rheology and reduce fluid loss; once the process is over, these polymers must be removed to allow for oil and gas production from the subsurface formations. Enzymes are used to break the polymer coat formed at the face of the formation. Oil and gas wells’ temperatures usually exceed 200 °F and can reach 400 °F, and this requires highly stable enzymes at elevated temperatures. The authors utilized galactomannanase (β-mannanase) and conducted a series of experiments at a wide range of temperature and enzyme concentrations. Through their experiments, they have found that galactomannanase can work better in ambient conditions. They also found that enzyme viscosity can be decreased without affecting the molecular weight of the fluid [13]. Researchers engineered the genetically modified β-mannanase enzymes that are thermally and pH stable. The enzymes that they had devised caused a significant reduction in fracturing fluid as compared to ammonium persulfate [14]. The following table (Table 1) shows the different applications of enzymes during drilling and completion operations.

## 3. Direct Evolution

Darwinian evolution is a natural process by which the selection of a particular character is favored by natural selection through environmental selection pressure. Genetic diversity, mutation, recombination, and error in DNA replication are driving forces to adopt the environmental selection pressures. This leads to the development of specific traits through natural selection required for the survival of species in specific extreme conditions—in the hottest and coldest environments [26]. These traits are expressed in the form of proteins, enzymes, or other specific characters to mitigate harsh environmental conditions. Enzymes from such living organisms are isolated and then used at the industrial level for multiple purposes such as feed, food, medicine, etc. [27,28]. These selected traits are passed to successive generations. However, evolution is a time-consuming process. It takes much time (sometimes millions of years) to develop a specific trait for a particular environment. In recent decades, researchers have mimicked the natural evolution process at the lab bench, known as direct evolution, which is a method that was awarded the Nobel Prize in chemistry in 2018—it is a specific conceptual and methodological approach within enzyme engineering. The four steps (Figure 2) involved in in vitro evolution or direct evolution are: (1) creating diversity in enzymes through different mutagenesis methods such as random mutagenesis or site direct mutagenesis in the initial gene pool; (2) making libraries of mutant/variant types of enzymes through recombination; (3) separation of desired enzymes through applying evolutionary pressure, high throughput screening for variants with desired properties; (4) amplification of desired variants to repeat the further cycle of mutagenesis, library formation, screening, and finally amplification. This cycle is repeated multiple times until the enzymes gain desired characteristics [29]. In direct evolution, diversification and screening are repeated multiple times. Different techniques are employed to develop diversity in enzymes. Error-prone polymerase chain reaction (epPCR) is one of the most efficient strategies to develop the desired traits/characteristics in enzymes. The direct evolution method gradually accumulates the particular advantageous mutation. Low error-rate mutagenesis, such as 1 to 2 amino acids changes in the protein by epPCR, can avoid the missing of beneficial mutations [29].

### The Need for Direct Evolution to Increase the Thermostability of Enzyme Breakers

Industry demand for enzyme breakers is high in order to work under extreme environmental conditions such as high temperature and high pH. There are many examples of thermostable enzymes produced by direct evolution for gel breaking in the oil industry. Mannanase enzymes were isolated from a hydrothermal vent and direct evolution was later on performed to improve their thermostability. The mannanase enzyme is widely used at the industrial level such as in feed, medicines, and in the upstream oil industry as a gel breaker. Mannanase breaks the β-1,4-glycosidic bonds in guar gum with fewer residues [11]. As compared to others gel breakers, the mannanase enzyme was first isolated from hot spring bacteria to work in extreme oil wells at high temperatures. However, the temperature stability requirement of the mannanase enzyme was higher to work efficiently in upstream oil industry such in extreme conditions of oil wells. In recent decade direct evolution strategy has been adopted to increase the thermostability and catalytic efficiencies traits of mannanase enzyme. The mutant type (m RmMan5A) with enhanced thermostability was generated through direct evolution from wild-type β-mannanase (RmMan5A) isolated from fungus, *Rhizomucor miehei* [30]. The catalytic activity of mutant (mRmMan5A) increased three-fold under acidic and thermophilic environmental conditions. After genome sequence confirmation, three amino acids (Figure 3) (Tyr233His, Lys264Met, and Asn343Ser) were substituted in the mutant (mRmMan5A) type. These three mutations increased the thermostability of the mutant variant. After comparative site direct mutagenesis, it was confirmed that the Tyr233 hydrophobic residue was substituted with histidine (His), That changed its surface nature from hydrophobic to hydrophilic which ultimately enhanced the stability. In Lys264Met variants, the positively charged residue at Lys 264 was changed to non-polar residue Met that increased the catalytical efficiency of Lys264Met. According to a structural analysis of mutant enzyme (RmMan5A), it was confirmed that the weak negative surface charge was changed to a strong negative surface charge through direct evolution that increased the thermal stability of the β-1,4-glycosidic mutant variant.

During direct evolution, wild-type mannanase from *Thermotoga maritima* sp. was employed for mutagenesis pipelines. Through a combination library, a single-site mutation was constructed using recombination and evolution technology; across 240,000 screened clones, 16 single mutations were found to be effective in mutants to increase thermostability compared to wild-type clones. From temperature and pH profiling studies, it was confirmed that the mutant variant can work in a broad range of temperature (80 to 275 °F) and pH to break borate cross-linked guar effectively with a small dose (less than 100 ppm) in the upstream oil/gas industry [14]. Recently, through the saturation mutagenesis technique of direct evolution, a 3.6-fold thermostability was extended by the generation of advanced-type recombinant mutant (ManM3-3, ManM5-10) of β-mannanase which was isolated from the thermophilic Gram-negative bacteria, *Thermoanaerobacterium aotearoense* sp. *(*Figure 4, [31]). During this process, iterative saturation mutagenesis was applied to generate 7000 clones; finally, after screening at high temperature, four clones: ManM3-3 (D143A), ManM4-17 (S53D, G56H), ManM4-19 (S53L, G56G), and ManM5-10 (E32K, E34S), were selected. In further characterization against high temperature, two clones: ManM3-3 (D143A) and ManM5-10 (E32K, E34S), performed well. However, in the analysis, the mutant ManM3-3 (D143A) outperformed the others. The substitution of a hydrophilic amino acid (ASP) to a hydrophobic amino acid (Al) in the *β*-sheets enhanced the internal hydrophobic interaction that ultimately increased the thermal stability of the variants ManM3-3 and ManM5-10.

Cellulase enzymes degrade cellulose to glucose. Cellulase includes enzymes such as β-1,4-endoglucanase, cellobiohydrolase, and β-glucosidase. Cellulase can degrade the β-1,4-glycosidic bonds in guar gum efficiently with fewer residues. Many studies show that the working optimum temperature of cellulase enzyme is between 45 to 60 °C. However, this temperature does not meet the requirement of food, feed, and upstream oil industries for multiple purposes. Researchers used the error-prone PCR and DNA shuffling techniques for direct evolution of β-glucanase isolated from bacteria, *Bacillus subtilis*, to increase the thermostability for industrial usage. With direct evolution, they generated two mutants, EGsl and EGs2, which can hydrolyze polysaccharides at 149 and 153 °F, respectively, which exhibit 37 and 41 °F higher working temperature than the wild type (112 °F) [31]. Recently, direct evolution has been employed to increase the thermal stability of β-glucosidase with random mutagenesis and high-throughput screening techniques. The mutant generated through direct evolution contained the two mutations, A17S and K268N, which increased their catalytic activity and thermostability. Through genome sequencing and multiple local alignments at residue number 17, the alanine residue was replaced by a serine residue that increased their residual volume which changed the higher thermostability in the generated mutants [33]. There are many other examples of using the direct evolution to increase the thermostability of enzymes (Table 2), including galacto-N-biose/lacto-Nbiose I phosphorylase (GLNBP), amyl phytase,endo-b-1,4xylanase (XynA), tyrosine phenol-lyase, and Β-glucosidase. The galacto-N-biose/lacto-Nbiose I phosphorylase GLNBP is a crucial enzyme for producing and promoting intestinal bacteria growth in the human body, and this enzyme has many industrial applications [34]. Two single mutations were selected through direct evolution by creating random mutagenesis; the first generation of the GLNBP enzyme increased the 50 °F temperature stability in the mutant than the wild types. A further double mutation was generated in the subsequent generations that enhanced the thermal stability up to 68 °F than the wild type, which increased the industrial production of GLNBP. Through structural analysis of the protein, it was hypothesized that the mutation of D576 increased the hydrophobic residues and the residues increased the thermostability of the variants [35]. Amylase has extraordinary significance in the microbial biotechnology industry with approximately 25% of the world’s commercial enzyme market share. The amylase enzyme has wide application in the food, fermentation, textile, paper, pharmaceutical, and oil industries as a gel breaker [34,36]. The high thermal stability is key in the wide application of amylase at the industrial level. The thermal stability and catalytic activity of amylase enzyme strain IM6501 (ThMA) were enhanced through the direct evolution method/DNA shuffling. A total of seven single mutations (R26Q, S169N, I333V, M375T, A398V, Q411L, and P453L) were accumulated in the amylase high thermostable mutant, ThMA-DM, after four rounds of direct evolution. These accumulated mutations increased the 59 °F in thermal stability as compared to the wild type. It was observed that the seven substitutions (Phe, Tyr, Lys, Val, Ser, and Thr substituted with Met375) increased the thermal stability but dramatically decreased the enzymatic activity [37]. Xylan is the most abundant hemicellulose, which is composed of lignocellulose biomass. The endo-B-1,4-xylanase (XYnA) can degrade the Xylan, but its function is limited by high temperatures. This limitation was overcome by creating the thermoresistant endo-B-1,4-xylanase (XYnA) enzyme from *Thermomyces lanuginosus* fungus through direct evolution using the error-prone PCR method. The first generation of this enzyme through mutagenesis and screening showed the three mutants (2B7-6, 2B7-10, 2B11-16) with enhanced thermal activity and stability. The second generation developed through direct evolution showed enhanced thermal stability and catalytic activity. In this generation, 2B7-10 mutant stability or thermal resistance outperformed compared to the wild type and other strains [38]. The phytase enzyme is used significantly in the feed industry to increase the bioavailability of nutrients for animal digestion. In the feed production industry, the feed pletting procedure operates at a high temperature that causes a reduction in the working efficiency of the phytase enzyme [39]. In one study, researchers increased the thermal stability of phytase through direct evolution by using error-prone PCR and DNA shuffling methods. The three mutants (E156G, T236A, Q396R) produced in the first generation by direct evolution exhibited 84% higher catalytic activity. The second generation produced through PCR and DNA shuffling increased the thermal stability of three mutants [40] 3,4-Dihydroxyphenyl-L-alanine (L-DOPA) is used as a drug against parkinsonian neurodegenerative disease worldwide [41]. To improve the thermal stability, 25 TLP mutants were generated through direct evolution, from which two mutants (E313W and E313M) showed higher thermostability and yield of L-DOPA. To improve the thermal stability, 25 TLP mutants were generated through direct evolution, from which two mutants (E313W and E313M) showed higher thermostability and yield of L-DOPA [42,43].

In the above mentioned, all examples of engineered thermostability enzymes through direct evolution techniques such as site direct mutagenesis, DNA shuffling, error-prone PCR, and gene recombination employed to create diversity, followed by selection, and amplification, has been proven a powerful tool to increase the thermostability of enzymes by accumulating multiple beneficial mutations, during which a large number of mutants are created to reach the desired level of changes. Now, with advanced screening strategies, one can screen billions of mutants created by direct evolution. There is immense diversity, even for small-sized enzymes, and there is great value to be discovered with direct evolution. The same above-mentioned direct evolution biological protocols could be adapted to further increase the temperature resistance and thermal stability of enzyme breakers to degrade the guar gel viscosity during hydraulic fracturing in the upstream oil and gas industry for efficient oil and gas extraction from the reservoir.

## 4. Rational Design

The enhancement of enzyme activities can be achieved through different types of rational design approaches (Figure 5, Table 3)**.** In the last couple of years, computation approaches have been successfully used to design the enzymes to tolerate high temperatures and pressures for multiple industrial applications. The first time the rational design strategy was used to design cytokine enzyme variants to improve their thermostability was in 2002 [45]. Improving the stability of enzymes at high temperatures, carried out by enzyme engineers, is currently a demanding characteristic. The active site′s geometry and structure give a clue for the development of maximum catalytic activity and thermal stability of enzymes Therefore, computational strategies have been used to change the backbone conformational changes and fold/unfold changes in enzymes. Recently, several in silico methods have been designed based on a specific mutation in the genome and their effect on the stability of enzymes under high temperatures. However, the reliability of such types of in silico methods is still unreliable. All the rationally designed techniques used to increase enzyme temperature stability are also based on phylogenetic analysis. Genes are compared to homologs based on BLAST or MSA methods. Usually, the phylogenetic analysis discourages non-beneficial residues and encourages beneficial residues in the desired thermostable-enhanced enzymes The ionization of charged particles, optimization of the loop, and free energies used as critical factors for the development of high-temperature-stable enzymes and free energies of enzymes have also helped in computational-based rational design development for the increment of enzyme thermal stability [46].

There are several currently available rational design strategies to improve enzyme thermal resistance; the most commonly used rationally designed technique is protein or enzyme engineering through disulfide bridges, surface charge optimization, hydrogen bond, proline factor, and salt bridge [52]. The disulfide bridge plays an important role in the structural stabilization of the enzyme with a reduction in unfolding entropy. Different molecular simulations can be employed to design thermostable enzymes based on disulfide bridges. In surface charge optimization, many charged amino acids are present at the enzyme surface; by balancing or optimizing charges through dynamic simulation, users can computationally design thermostable enzymes. The B-factors in the structure of proteins have a direct relation with diffusion, the higher the B-factors means higher diffusion and less stability. All these strategies can increase enzyme stability by increasing protein foldability, rigidity, and decreasing enzyme subunits’ free energy at the structural level. In many applications of rational design, increasing the number of hydrogen bonds and salt bridges increases the thermal stability of many enzymes for industrial and pharmaceutical applications. Proline is the most important factor for increasing the stability of enzymes [53]. The introduction of proline into the peptide chain’s loops increased the thermostability of many enzymes. To increase the thermal stability of endo-1,4-β-galactanase (TSGAL), a single proline substitution—K31P, T172P—is performed. Proline factors are also employed to increase the thermostability of green fluorescence protein (GFP) and collagen. The hydrophobic core is very important for increasing the foldability of enzymes to develop tolerability against physical factors. Researchers are introducing different types of computationally based strategies to develop a fully hydrophobic compact core of enzyme against high temperature [54,55]. Enzyme stability can be improved through the introduction of many mutations. However, during rational design, sometimes a single mutation can destabilize and disturb the enzyme protein folding as well as enzyme thermostability with a low boiling point. That is why a high amount of predictive precision is essential for the rational design of enzymes. Unfortunately, all existing approaches have a very high possibility of negative mutagenesis inside the enzyme coding gene. That is why enzyme stability can be maintained by adding predictive stabilized mutations inside the enzyme coding gene [56].

### 4.1. Rational Designed Algorithms for the Engineering of Thermostable Enzyme Breakers

#### 4.1.1. FRESCO

The framework of rapid enzyme stability by computational libraries (FRESCO) was created developed in 2014 [57]. FRESCO contains five significant steps: (1) A library of mutations created through Rosetta and FoldX software in the first step; the mutations that give the enzyme’s stabilization effect are retained. (2) In the second step, the undesirable mutations are eliminated. (3) In the third step, screening of mutants’ libraries is performed through different dynamic simulations. This leads to an increase in the mean square. The molecular dynamic simulation is employed to evaluate enzyme efficiency based on the melting point and catalytic efficiency [58]. Next, the point mutation is validated experimentally based on the melting point and catalytic activity. (4) After library validation, the point mutation is obtained, tested, and verified in the fourth step. In the last fifth step, the validated point mutation is screened, simulated, and the final stabilized enzyme variant is added.

There are many advantages of FRESCO that employ different approaches to design thermostable enzyme breakers artificially in the oil industry. However, the two commonly significant benefits are that it creates different types of beneficial variants and uses different methods such as Rosetta to filter them. The second is that it can assess the enzyme efficiency through visual examination.

#### 4.1.2. PROSS

The enzyme stabilization algorithm, the protein repair one-stop-shop (PROSS), was developed in 2016 [59]. Firstly (1), a sequence is analyzed on the basis of homologous alignment. This algorithm technique removes the rare point mutations in the sequence on the basis of sequence specificity. (2) After the Rosetta software tool is employed to select those mutations that provide less energy, less energy compared with wild types gives more thermal stability. (3) The validated mutations with less energy are combined.

Based on less energy, the scoring of the algorithm is designed. The PROSS′s primary benefit is that the result’s analysis is reproducible with high efficiency in silico and in vitro for the production of thermostable enzyme breakers.

### 4.2. The Need for Rational Design to Increase the Thermostability of Enzyme Breakers

An improved thermostable β-mannanase enzyme mutant (mutant336 (A336 P) was generated through a rational design strategy (Figure 6). Firstly, the 3D structure of wild-type β-mannanase (ManTJ102) was generated through Protein Data bank (PDB), the flexible area was selected through B-factors. In the molecular dynamics simulation, annealing and heating were performed virtually to obtain the dynamic transition temperature (T_dtt_). Virtual mutation was employed to select the variable mutations computationally that improves the thermostability of wild-type β-mannanase (ManTJ102). During virtual mutation, the flexible areas of residues 330–340 were selected. Through wet lab mutation analysis, it was confirmed that the mutant336 with position Ala336pro is the most critical residue to increase the thermostability of wild-type β-mannanase (ManTJ102). Through rational design, it was confirmed that the half-life of mutant336 was 120 min at 60 °C (140 °F) which was 24-fold higher than the wild type. It was also used as a gel-breaking agent in the petroleum mining industry; a temperature of over 140 °F showed many advantages such as controllable breaking time, full hydrolysis, high-permeability recovery, and environmentally friendly performance [60]. The amylase enzyme has been employed as a gel breaker in the upstream oil industry individually or in a mixture of different gel-breaking enzymes. Thermostability is the main concern in the usage of amylase for the upstream oil industry. However, for the pharmaceutical industry, the rational design strategy was employed to increase the thermostability of amylase to enhance the production of maltose syrup in high-temperature environments. Through B-factor analysis and molecular dynamics simulation, seven mutations were identified with improved thermostability character. Out of seven, the optimum working temperature of three mutations (G128L/K269L/G393P) increased from 122 to 150 °F [47].

Rational design methodologies have been employed to design the cytosine deaminase enzyme against high temperatures. In this study, the free energy or energy function was used to evaluate cytosine deaminase enzyme efficiency to fix the target sequence in the amino acid chain. However, the adaptation of these sequences with low free energy was automatic. Based on this experiment, the author identified three beneficial mutations. After combining these mutations, they developed remarkable efficiency and thermostability of cytosine deaminase enzyme against high temperature [48]. Lipase is used in industries as a catalyst for the production of multiple commercial products. Industrial catalysis processes occur at high temperatures; rational design with computational modeling software such as the Rosetta approach is used to develop thermal resistance and stability in the lipase enzyme. This has increased the melting temperature up to 44 °F [49]. Cutinase is a key biocatalyst for polymer degradation. This enzyme is widely used in the cosmetics, food, and insecticide industries. The rational design with biological modeling software Rosetta was used to increase the thermostability of cutinase, increasing the thermostability 10 times higher than the wild type [50].

Through rational design, the thermal mechanism based on the structure and function of a large number of enzymes or proteins has been investigated. As in the above-mentioned examples, many factors (disulfide bond, surface charge, proline effects, B-factors) are interrelated for improving the thermostability of enzymes. These factors can be optimized computationally with less time as compared to direct evolution to increase the thermal stability of enzyme breakers to break the guar gum gel during hydraulic fracturing in the upstream oil and gas industry. The above-mentioned rational design strategies with their protocols can be adapted to increase the thermostability of enzyme breakers in the fracturing hydraulic process.

## 5. Semi-Rational Enzyme Engineering to Increase the Thermostability

Traditionally, direct evolution requires a million mutant libraries to generate the diversity to screen out the desired mutant. These approaches are time consuming and problematic due to large space requirements to accommodate the million mutants′ libraries. Traditional enzyme engineering approaches are also limited due to the biasness and genetic code degeneracy in the libraries. That is why there was a need for smaller, high-quality libraries that could lead to the development of semi-rational design strategies to engineer the enzymes. In semi-rational enzyme engineering, researchers can pre-select the enzyme target sites through employing computationally predictive algorithms based on structure and function prediction. This technique focusses on specific amino acid mutations with less library generation which dramatically reduces the size of the library. Through semi-rational sequence and structure-based design strategies, machine learning algorithms have become effective tools to engineer the enzymes. Together with these tools, the enzyme function and stability determined through predictive computational tools, followed by wet lab work, led to the mutation and screening of that particular substitution [51].

By employing semi-rational design approaches, researchers can design novel functions such as thermostability and catalytical activity of enzymes by tailoring the specific amino acids virtually and practically. In one study, the SCHEMA, a structurally guided recombination predictive model of semi-rational design, was employed to increase the thermostability of CBH II cellulases. Through this predictive model, the thermostability working temperature of cellulase chimeragenesis was increased 59 °F higher than optimum with high catalytical activity [61]. The sequence-specific structural-based mutation was employed to improve the thermostability of the chitinase enzyme. The specific mutants (S244C-I319C/T259P) were generated which showed a 26.3-fold increase in half-life at 122 °F with less libraries [62]. Through the semi-rational MD simulation predictive mode, the mutant had a stable active structure which was confirmed after generation of the mutation. The semi-rational MD simulation was employed to increase the thermal and catalytical activity of pyranose 2-oxidase enzyme. Through sequence analytical studies, it was confirmed computationally that the substitution of Glu-Lys at amino acid position 540 increased the enzymes’ thermostability and catalytic activity. Then, a specific mutant was generated that increased the temperature by approximatey 50 to 58 °F [63]. Recently, this technique was employed to enhance the thermostability of the heparinase 1 enzyme through computationally aided designs, molecular docking, and dynamic calculations. Firstly, through molecular dynamics, the single mutant (Q157H) with higher thermal stability was identified and then site-directed mutagenesis was employed to create the specific mutation. Compared with the WT, the half-life time of the mutant Q157H was 6.0-fold and 2.08-fold higher than that of the WT at 25 and 37 °C [64].

Using semi-rational designing predictive tools, researchers can generate novel thermostable enzyme breakers with fewer numbers of libraries and minimal resources. Firstly, the machine learning-based sequence to functionally based algorithms and MD simulations can be employed in gas and oil industry to generate specific novel enzyme breakers with higher thermostability to degrade the guar gum gel during hydraulic fracturing. The specific high temperature resistant enzyme breakers could be generate through combination of computationally with higher thermostability predictive chances that leads to wet lab specific mutations generation. 

## 6. Conclusions and Future Outlook

Recent discoveries in molecular science such as direct evolution and rational designs have been used to engineer highly thermostable enzymes for multiple industrial purposes. However, there are few studies available in the literature on increasing the thermostability of enzyme breakers through direct evolution in the upstream oil industry. These engineering techniques have increased the enzyme breakers’ working temperature up to 275 °F but the oil and gas wells’ temperature usually exceeds 200 °F and can reach 400 °F. Still, thermostability remains a challenging issue for its usage in the upstream oil industry. Enzyme engineering depends on tuning the catalytic site, changing the amino acid sequence, and altering the bond strength to increase enzyme stability for the thermostable production of enzymes through direct evolution and rational design There are a lot of recent insights in the direct evolution and rational design using machine learning. These approaches have increased the thermostability of different enzymes in the feed and medicine industry. These strategies can also be employed to increase the thermostability of enzyme breakers to degrade the thickness of hydraulic fluid in hydraulic fracturing processes. The introduction of protein structure prediction machine learning has increased the possibility of designing enzyme structures with desirable thermal stability characteristics or amino acids combinations.

The current main challenge of enzyme engineering technologies is a critical assessment of protein or enzyme structure prediction. In the future, with the help of quantum computing, researchers will be able to develop the tools necessary to solve this issue. It is expected that more and more research data will be utilized to advance direct evolution technologies and advance data-driven rational designing techniques to improve protein/ enzyme breaker stability against temperature and pH in biochemical or biological laboratories for their application in the upstream oil and gas industry.

## Figures and Tables

**Figure 1 ijms-23-01597-f001:**
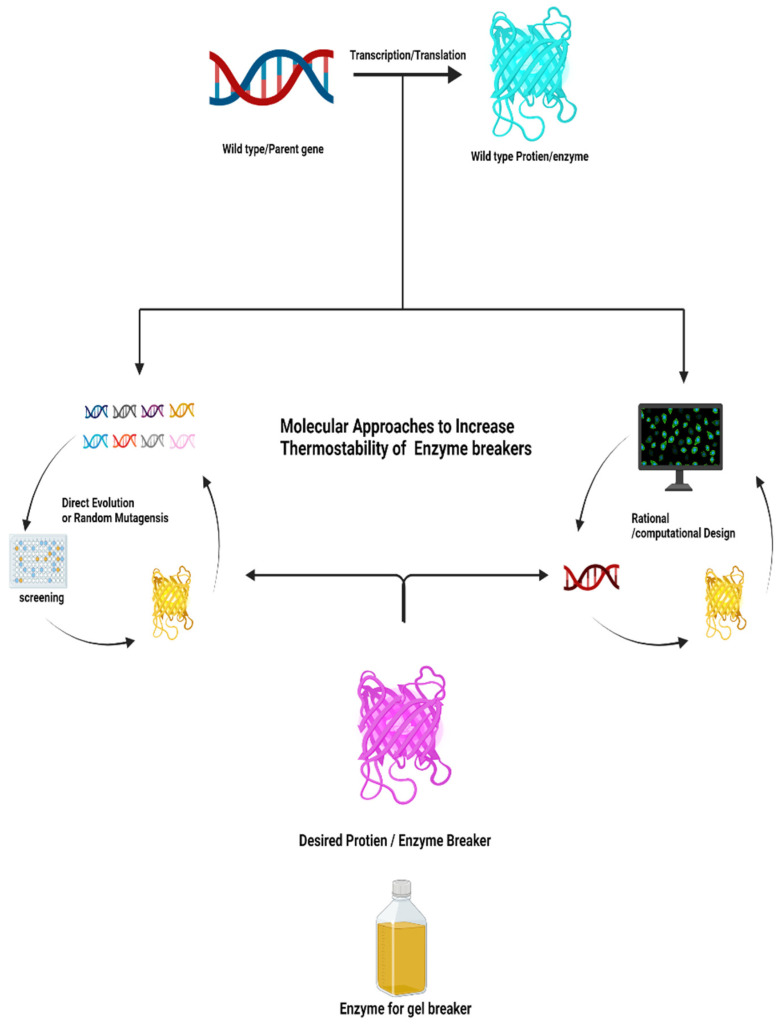
Abstract overview of molecular approaches used to increase the thermostability of enzyme breaker, created by https://biorender.com web tool.

**Figure 2 ijms-23-01597-f002:**
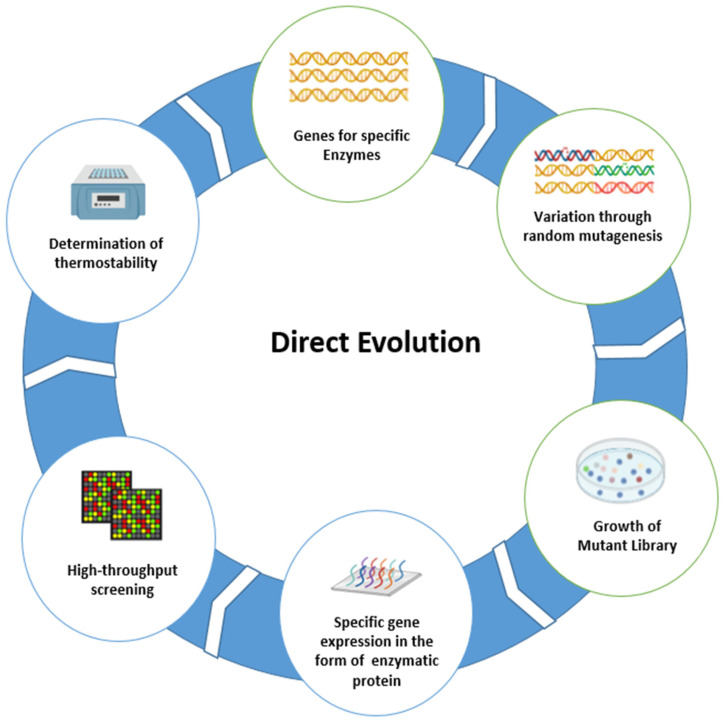
Schematic representation of direct evolution steps, created by https://biorender.com web tool.

**Figure 3 ijms-23-01597-f003:**
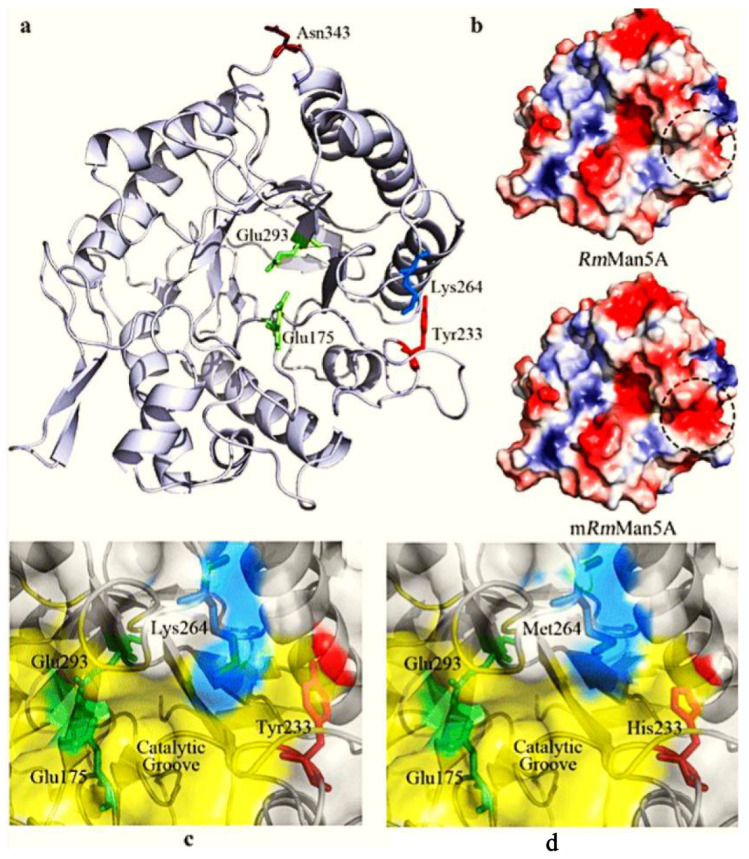
(**a**) Localization of three residues (Tyr233, Lys264, and Asn343), which has improved the thermostability of mutant mRmMan5A. (**b**) The protein surface charge of RmMan5A and mRmMan5A. The most negative and most positive electrostatic potentials are indicated by red and blue, respectively. (**c**) The location of Tyr233 and Lys264 in RmMan5A before mutation. (**d**) The location of His233 and Met264 in mRmMan5A after mutation. The active site and catalytic groove were colored in green and yellow, respectively [30].

**Figure 4 ijms-23-01597-f004:**
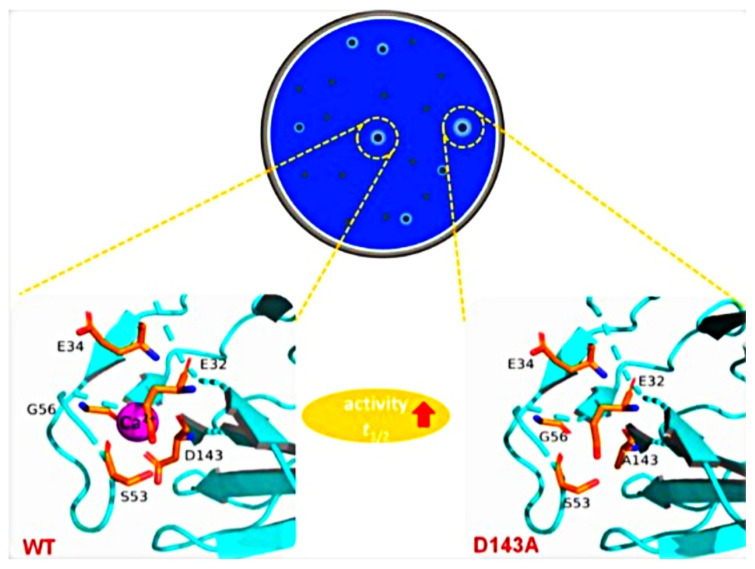
Mutant with enhanced thermostability generated through direct evolution [32].

**Figure 5 ijms-23-01597-f005:**
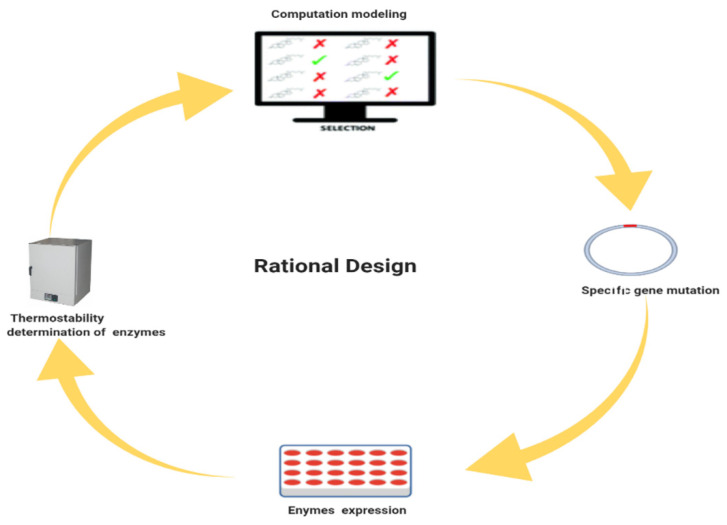
Schematic representation of the rational design, created by https://biorender.com web tool.

**Figure 6 ijms-23-01597-f006:**
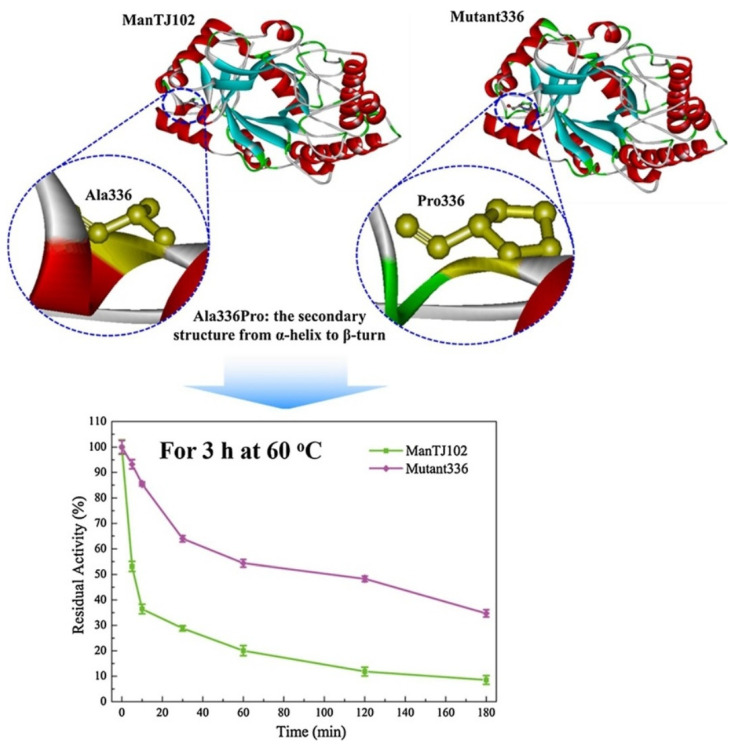
Thermostable β-mannanase enzyme mutant (mutant336 (A336 P) generated through rational design [60].

**Table 1 ijms-23-01597-t001:** Enzymatic applications in upstream oil and gas operations.

Application	Stabilizer/Enzyme	Notes	Ref
Filter cake removal	Chelant/enzyme	The system was stable and effective to break hydroxypropyl starch and xanthan gum	[15]
Fracturing fluid	Lignosulfonates/enzyme (mannanase)	Efficient breaking of guar-based fluid at high pH (10.5) and high temperature (160 °F)	[16]
Fracturing fluid	Polyelectrolyte complex nanoparticles (enzyme carrier)	Fracture conductivity enhanced compared to the old technique due to the efficient removal of HPG polymer	[17]
Well stimulation	Enzymes as polymer breaker	Enzymes showed better polymer clean up compared to conventional breakers (sodium persulfate) and better well productivity	[18]
Mud cake removal	a-amylase system (old) and structurally reinforced a-Helix system (new)	The old enzyme caused damage at 212 °F and precipitated. The new system effectively hydrolyzed the biopolymer at high temperatures.	[19]
Fracturing fluid	Enzyme breaker	Efficient cleaning of polymer and long-term sustained production from the well. Low temperature, 100 °F	[20]
Gravel pack clean up	Enzyme breaker	The polymer was cleaned and the plugging of the gravel was reduced	[21]
Polymer and breaker adsorption	Enzyme	Care should be taken when using real rocks saturated with oil compared to cleaned rocks. Oil may affect enzyme activity.	[22]
Drilling fluid damage	Enzyme	Starch was broken using polymer linkage-specific enzymes. This process reduced the formation damage due to polymer in the drilling fluid.	[23]
Fracturing fluid	High-pH tolerant enzymes (modified hemicellulose)	The enzyme remains active at up to pH values of 11.5. Crosslinked guar was broken using this enzyme at a low temperature (120 °F).	[24]
Crosslinked fluids	Enzymes versus oxidative breakers	Enzyme breakers yielded better and homogenous breaking compared to oxidative breakers	[13]
Filter cake removal	linkage specific enzymes	The modified enzyme showed better polymer breaking compared to a mixture of generic hydrolytic enzymes. Amylases, cellulases, and glucosidases enzymes were very efficient in breaking the modified crosslinked starch in the filter cake during drilling operations.	[25]

**Table 2 ijms-23-01597-t002:** Engineered thermostable enzymes are generated through direct evolution.

Enzymes	Substrate	Applications	Findings	Ref
Mannanase enzyme	β-1,4-glycosidic bonds	Upstream oil, Feed and medicine industry	Through direct evolution three mutations (Tyr23His, Lys264Met and Asn343Ser) were identified that, has increased their catalytic and thermostability. Through iterative mutagenesis, out of 240,000 clones isolated one mutant (ManM3-3) outperformed under high temperature.	[30,31]
Cellulase	β-1,4-glycosidic bonds, Cellulose	Enzyme breaker or Gel breaker in oil Industry, feed, textile Industry	With direct evolution two mutants, EGsl and EGs2 were generated with enhanced thermostability at 149 °F and 153 °F respectively, which exhibit 37 °F and 41 °F higher working temperature than wild type (112 °F).	[33]
Galacto-N-biose/lacto-N-biose I phosphorylase (GLNBP)	Lactose- N-biose I	Milk Industry, Food industry	That study increased the thermostability of GLNBP enzyme by 68 °F by creating the C236Y and D576V mutants.	[44]
Amylase	Carbohydrates	Food, textile and paper/pulp, Baking Industry	In this study total 7 single gene mutations were accumulated in thermostable mutant after four rounds of direct evolution.	[37]
Endo-B-1, 4-xylanase(XYnA)	Xylan	Biofuel production and pharmaceutical industry	Their study increased the thermostability of XynA enzyme, the four mutants’ exhibit higher thermal stability in the first generation produced through random mutagenesis. The 2B7-10 mutant produced in second generation out- performed than the wild type.	[38]
Phytase	Phytate	Feed industry	Three mutants were generated in first generation through direct evolution which has 84% higher activity than wild type. The subsequent second generation has increased the thermal stability.	[40]
Tyrosine phenol-lyase	Tyrosine	Pharmaceutical industry	The 25 mutants were generated through direct evolution, among these only two shows the higher thermostability than other.	[42]

**Table 3 ijms-23-01597-t003:** Engineered thermostable enzymes generated through rational design strategy.

Enzymes	Substrate	Applications	Findings	Ref
β-mannanase		The upstream oil industry, feed, and medicine	The mutant (mutant336 (A336 P) with enhanced thermostability generated through rational design.	[47]
Amylase	Maltose	The upstream oil industry, feed, and medicine	The three generated mutations (G128L/K269L/G393P) through rational design have increased from 122 °F to 150 °F.	[48]
Cytosine deaminase	Cytosine	Anticancer	The three thermostable mutations were identified through rational design with Rosetta software that has increased the melting temperature (Tm) of enzyme up to 50 °F.	[49]
Lipase B	Triglycerides	Pharmaceutical and food industry	The rational design with computational modeling software Rosetta, that increases the Tm up to 45 °F.	[50]
Cutinase	Degrade the cuticle polymer	Pharmaceutical industry	The rational design has significantly improved the thermostability by 10-fold higher than the wild type.	[51]

## Data Availability

Not applicable.

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
