# Peer review of "A Review of Advanced Molecular Engineering Approaches to Enhance the Thermostability of Enzyme Breakers: From Prospective of Upstream Oil and Gas Industry"

_ijms, 2022, doi:10.3390/ijms23031597_

Round 1

Reviewer 1 Report

  • Line 66 page 2 Introduction: what is the advantage to use a « human » enzyme? It is not clear.
From which organism come the traditionnal enzyme used to improve the process of hydraulic fracture?
  • add a « , » page 2 Line 68 « as compared to oxidizers, enzymes »
  • Page 2 Line 73 replace « gell » by « gel »
  • Page 7 Line 167 replace « mannanses» by « mannanase »
  • Page 7 Line 168 replace « libarary » by « library »
  • Table 2 and Table 3. The authors repeat inside the two tables many times « the protocole can be adapted » or « can be employed » « can be used » « can be implemented ». Could it be possible to write few lines to explain how to use those protocols for the application of interest and do not put it inside the tables.
  • Page 12 Line 291 could the authors give some examples of thermostability improvement after introduction of a proline in a sequence.
  • Title? The title do not seem completely correct. It is not a review with example of success of improvement of thermostability of enzyme breakers in upstream oil and gas industry. But it is a review about genetic or bioinformatic methods to improve in general enzyme thermostability. In a second step, those methods could be employed for the application of interest. As suggested in the comment about the table, it could be great to add text in discussion to do the link between the methods described and listed in the review and how they could be used for upstream oil and gas industry. The screening have certainly to be well designed.

Author Response

Dear Prof. Dr. Sergio F. Sousa,  

Editor-in-Chief of IJMS,

Thank you very much for giving us an opportunity to revise our manuscript (ijms-1539599). Taking into consideration the Editor-in-Chief and Reviewers Comments, we have thoroughly checked and revised the final version of our manuscript. Also, the manuscript has been corrected by an English expert for the editorial corrections. The corrections are highlighted with yellow color in the revised manuscript. We have done a little medication, we added one more corresponding author ( Prof. Dr. Ajmad Bajes Khalil). We hope that our revised manuscript will be accepted for publication in the reputed IJMS journal. Please find below the point-by-point response of the questions raised by the Reviewers.

Thanks

Point by point Response to the Comments:

REVIEWER No. 1

Comment: Line 66 page 2 Introduction: what is the advantage to use a « human » enzyme? It is not clear.

.

Response: Thank you very much for your comment, we changed it accordingly at the line number 63 of revised version.

Comment: From which organism come the traditional enzyme used to improve the process of hydraulic fracture?

Response:  Thanks for your comment. We have written it accordingly from the line number 69 to 71.

Comment: add a « , » page 2 Line 68 « as compared to oxidizers, enzymes »

Response:  Thanks for your suggestion. We have changed it accordingly at the life number 71. Thanks

Comment: Page 2 Line 73 replace « gell » by « gel »

Response: Thanks for critical review. We have changed it accordingly at the line number 75 of revised manuscript.

Comment:  Page 7 Line 167 replace « mannanses» by « mannanase »

Response: Thank you very much for your critical review. We have changed it accordingly at the line number 152.

Comment: Page 7 Line 168 replace « libarary » by « library »

Response: Thanks for your suggestion. We have changed it accordingly at the line number 179.  

Comment: Table 2 and Table 3. The authors repeat inside the two tables many times « the protocole can be adapted » or « can be employed » « can be used » « can be implemented ». Could it be possible to write few lines to explain how to use those protocols for the application of interest and do not put it inside the tables.

Response: Thank you very much for your excellent comment. We have changed it accordingly from line number from line number 264 to 275 for direct evolution and from line number 404 to 412 in revised manuscript.

Comment: Page 12 Line 291 could the authors give some examples of thermostability improvement after introduction of a proline in a sequence.

Response: Thanks for your suggestion. We have written the examples related to proline introduction to increase the thermostability from line number 318 to 321.

Comment: Title? The title do not seem completely correct. It is not a review with example of success of improvement of thermostability of enzyme breakers in upstream oil and gas industry. But it is a review about genetic or bioinformatic methods to improve in general enzyme thermostability. In a second step, those methods could be employed for the application of interest. As suggested in the comment about the table, it could be great to add text in discussion to do the link between the methods described and listed in the review and how they could be used for upstream oil and gas industry. The screening have certainly to be well designed.

Response: Thank you very much for your excellent suggestion. Yes, as you know in enzyme engineering we mutate the gene and then screen the desired protein with respective to their required trait.  As you know we are working with the important molecules of life (DNA, proteins), and we use the molecular tools (gene cloning, restriction digestions, ligation, cloning, CRISPR), computational tool (in silico softwares) and protein purification tools (chromatography, centrifugation) to engineer the molecules of lives.  That’s why we proposed our title with aligned to molecular engineering. Thanks

Reviewer 2 Report

The author may add the enzyme thermostability issue in oil and gas application into the introduction part and move the

The authors introduced the basic concept of directed evolution and rational design and also provided some successful examples. However, the authors need to further summarize the effects of those mutations (why those mutations can improve thermostability, by increasing the rigidity of overall structure or by changing the local conformation? Or some residues in a specific position can improve enzyme stability?). Maybe the author should provide more information for readers who are also interested in the engineering of enzyme thermostability.

Increasing thermostability is sometimes accompanied by a decrease in activity. Can the authors find any example about recovering the activity by new mutations after thermostability engineering?

Some researchers have successfully combined the high-throughput screening (directed evolution) and rational design to improve the enzyme performance including thermostability and the author should discuss it.

Figures 3 and 4. The font, size, and style are not consistent at all and the author can prepare the figure by themselves (using Pymol or discovery studio).

Author Response

Dear Prof. Dr. Sergio F. Sousa,  

Editor-in-Chief of IJMS,

Thank you very much for giving us an opportunity to revise our manuscript (ijms-1539599). Taking into consideration the Editor-in-Chief and Reviewers Comments, we have thoroughly checked and revised the final version of our manuscript. Also, the manuscript has been corrected by an English expert for the editorial corrections. The corrections are highlighted with yellow color in the revised manuscript. We have done a little medication, we added one more corresponding author ( Prof. Dr. Ajmad Bajes Khalil). We hope that our revised manuscript will be accepted for publication in the reputed IJMS journal. Please find below the point-by-point response of the questions raised by the Reviewers.

Thanks

REVIEWER No. 2

Comment: The authors introduced the basic concept of directed evolution and rational design and also provided some successful examples. However, the authors need to further summarize the effects of those mutations (why those mutations can improve thermostability, by increasing the rigidity of overall structure or by changing the local conformation? Or some residues in a specific position can improve enzyme stability?). Maybe the author should provide more information for readers who are also interested in the engineering of enzyme thermostability.

Response: Thanks for excellent comment and suggestions: we have edited our revised manuscript accordingly from line number 167 to 171, 194 to 197, 14 to 217, 226 to 228, 237 to 240. Thanks

Comment: Increasing thermostability is sometimes accompanied by a decrease in activity. Can the authors find any example about recovering the activity by new mutations after thermostability engineering?

Response: Thanks for your excellent comment and suggestion we have edited. Please, check at the line from 237 to 240. Thanks

Comments: Some researchers have successfully combined the high-throughput screening (directed evolution) and rational design to improve the enzyme performance including thermostability and the author should discuss it.

Response: Thanks for your comment and suggestion. We have edited it accordingly from line number 414 to 446.

Comment: Figures 3 and 4. The font, size, and style are not consistent at all and the author can prepare the figure by themselves (using Pymol or discovery studio).

Response: Thanks for your suggestion, we have tried our best to present in original published form with high resolution with reference. Thanks

Comment: The author may add the enzyme thermostability issue in oil and gas application into the introduction part and move the.

Response: Thanks for your excellent comment and suggestion. We have edited it accordingly from the line number 83 to 89. Thanks

Round 2

Reviewer 1 Report

The text needs to be editing. "mannanse" "themrostable" "protien"

The authors improve the manuscript and take into account the remarks and change the tables.  

Author Response

Comment: The authors improve the manuscript and take into account the remarks and change the tables.

Response: Thank you very much for your feedback. Thanks

Comment: The text needs to be editing. "mannanse" "themrostable" "protien".

.

Response: Thank you very much for your comment, we changed it accordingly at lines number 153,179,228, 308, and 312 of the revised version.

Reviewer 2 Report

The authors have already addressed most questions raised by reviewers. However, the figures have not been revised at all. Please unify the font style and improve the quality of these figures (especially figure 3 and 4).

The title should be revised to meet the requirement of review articles.

The authors should check if these very recent publications such as 1. 10.1080/10242422.2021.1976757, 2. 10.1007/978-1-0716-1826-4_9, 3. 10.1002/cbic.202100431 have been discussed and cited properly.

Author Response

REVIEWER No. 2

Comment: The authors have already addressed most questions raised by reviewers. However, the figures have not been revised at all. Please unify the font style and improve the quality of these figures (especially figure 3 and 4).

Response:  Thank you very much for your feedback. We have enhanced the quality of Figures 3 and 4 in the current version. Thanks

Comment: The title should be revised to meet the requirement of review articles.

Response: Thank you very much for your suggestion, we have changed it accordingly. Thanks

Comment: The authors should check if these very recent publications.

Response: Thank you very much for your suggestions. We have updated it accordingly from line number 448 to 453. Thanks
